# Work minimization accounts for footfall phasing in slow quadrupedal gaits

**James R Usherwood\*, Zoe T Self Davies**

Structure and Motion Lab, The Royal Veterinary College, Hatfield, United Kingdom

**Abstract** Quadrupeds, like most bipeds, tend to walk with an even left/right footfall timing. However, the phasing between hind and forelimbs shows considerable variation. Here, we account for this variation by modeling and explaining the influence of hind-fore limb phasing on mechanical work requirements. These mechanics account for the different strategies used by: (1) slow animals (a group including crocodile, tortoise, hippopotamus and some babies); (2) normal medium to large mammals; and (3) (with an appropriate minus sign) sloths undertaking suspended locomotion across a range of speeds. While the unusual hind-fore phasing of primates does not match global work minimizing predictions, it does approach an only slightly more costly local minimum. Phases predicted to be particularly costly have not been reported in nature.
DOI: https://doi.org/10.7554/eLife.29495.001

## Introduction

An understanding of the factors determining selection of gait parameters allows insight into what 'matters' to an animal as it locomotes. Many details of bipedal walking mechanics can be understood from the perspective of simple mechanical work minimization (*Kuo, 2002*; *Srinivasan and Ruina, 2006*; *Srinivasan and Ruina, 2007*); however, the situation for quadrupeds is less clear. While quadrupeds, like bipeds, tend to adopt 'symmetrical' walking gaits at low speeds, with left and right sides operating evenly in turn, a range of phases between hind and fore feet can be observed. The duration a foot spends in contact with the ground, expressed as a proportion of the stride cycle, is termed a 'duty factor' DF; walking gaits require a duty factor above 0.5, meaning that at least one foot of a pair is on the ground at all times. We follow Hildebrand's (*Hildebrand, 1965*) definitions for phasing of fore and hind limbs, defining phase as the 'proportion of stride interval [period] that footfall of forefoot follows hind on same side'. Using this convention, a typical horse walking with a 25% phase would have an even timing between each foot contact: starting with the left hind, next would follow the left fore, then right hind then right fore. A 'trot' footfall pattern, with diagonal hind and fore feet striking the ground at the same time would have a 50% phase; a 'pace' pattern with simultaneous ipsilateral feet contact (i.e. both left then both right) could be expressed as a 0% or 100% phase.

The aim of this paper is to provide an account for the ranges of hind-fore limb phasing observed across walking quadrupeds.

Stability-based models usually assume that footfall patterns should be favoured that allow the centre of mass to be supported – or most nearly so – above feet forming a 'polygon of support' on the ground. The unsatisfactory or incomplete nature of accounts for phasing based in terms of only static stability has been widely identified (*Gray, 1944*; *Jayes and Alexander, 1980*; *Vilensky and Larson, 1989*; *Cartmill et al., 2002*). To highlight some of the issues of stability-based approaches, it should be noted that:

- at DF <0.75, no footfall timing is capable of allowing continuous progression of the center of mass above a polygon of support;

**\*For correspondence:**
jusherwood@rvc.ac.uk

**Competing interests:** The authors declare that no competing interests exist.

- at DF >0.75, footfall phasings can be calculated that allow this (*Gray, 1944*; *McGhee and Frank, 1968*), but these phases are not adopted (even in tortoises! [*Jayes and Alexander, 1980*]);
- bipedal animals are capable of walking, and aspects of the mechanics of bipedal walking economy are precisely because the center of mass is allowed to vault passively without being maintained over a polygon of support;
- running gaits are not precluded due to periods of zero 'polygon of support'.

Some or all of these issues have been acknowledged to a greater or lesser extent, at least since Gray (*Gray, 1944*); however, the view persists that 'other things being equal, it seems reasonable to expect that moving animals would deploy their limbs in such a way as to maximize their support polygons over the stride period' (*Cartmill et al., 2002*). Stability-based accounts continue to dominate the interpretation of functional significance of limb phasing in walking quadrupeds (*Patrick et al., 2009*). Here, we account for the limb phasing of walking quadrupeds by developing a general numerical model to determine the implications of phasing on mechanical work demands (i.e. showing why other things are *not* equal), and provide simple geometric explanations for the model results.

To summarize the phenomena we wish to account for, we plot (*Figure 1*) published trends for primates and sloth, and values for a diverse range of species (52 species from 120 observations) from freely available YouTube clips and video from the Structure and Motion lab from other projects (*Table 1*, *Supplementary file 1*). The relationships should be uncontroversial (see *Supplementary file 1*): many species have been studied previously, and we do not claim significant novelty in reporting duty factors and phases for walking. Hildebrand's pioneering and thorough work in this area should be emphasized; the new kinematic data in this paper are intended to be illustrative and not comprehensive. Formal statistical approaches incorporating phylogenetic comparative methods are beyond the scope of this study. Clip selection criteria were: a clear view of all feet during unconstrained, apparently 'natural' walking for a complete stride cycle over approximately flat, level ground. An observation of a mouse and a hamster walking on a treadmill (see [*Spence et al., 2013*] and acknowledgements) is also included. A summary of literature findings is included in *Supplementary file 1*, which broadly agree with the measurements reported here. We have no statistically based view as to whether our data should be viewed as continuous or clustered (though we do show a regression line, and cluster into Groups using K-means assuming 2 groups); we proceed here by describing low and high duty factor species groupings as discrete for convenience only.

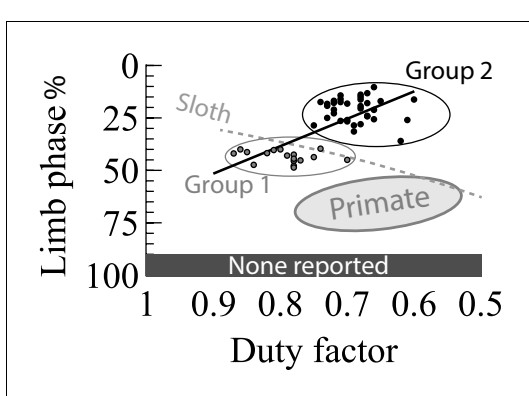

**Figure 1.** General relationships for walking quadrupeds between duty factor and limb phase following Hildebrand's conventions for symmetrical gaits: duty factor defined as the proportion of the stride cycle in which a hind limb is in contact with the ground; limb phase the proportion of the cycle (as a %) after initial hindlimb contact that the forelimb of the same side makes contact. Sloth relationship from (*Nyakatura et al., 2010*); Primate from (*Cartmill et al., 2002*). Points show values for a diverse range of quadrupedal species (*Table 1*), tentatively classified into two groups using K-means: Group 1 – high duty factor, phase 40–50% including reptiles, amphibians, mouse and hippo (grey points); Group 2 – the 'normal mammalian' grouping, with lower duty factor and phase around or somewhat below 25% (black points). The regression line relating to the presented points is: phase (%)=130 DF – 66. Phases greater than 80% are rarely if ever observed in steady, level gaits.
DOI: https://doi.org/10.7554/eLife.29495.002

## Group 1: Duty factor near 0.8

Group 1 animals (*Figure 1*, *Table 1*) walk with a duty factor around 0.8 and phase between 40% and 50%. This group consists of 11 species of reptile, 3 amphibians and 3 mammals. Members of this group might be linked by being relatively slow (e.g. hippo), slow-muscled ('cold-blooded' – consider tortoise muscle properties [*Woledge, 1968*]) or very small (mouse). If so, their high duty factors might be considered a strategy to limit the muscle activation cost due to instantaneous power demands (see

**Table 1.** See **Supplementary file 1** for further information.

| | Median duty factor | Median phase (%) | N | SD duty factor | SD phase (%) |
|---|---|---|---|---|---|
| **Group 1** | | | | | |
| Hamster* (*Mesocricetus auratus*) | 0.70 | 45 | 1 | | |
| Giant Salamander (*Andrias japonicus*) | 0.74 | 40 | 1 | | |
| Iguana (*Iguana iguana*) | 0.75 | 44 | 2 | 0.126 | 6.6 |
| Komodo Dragon (*Varanus komodoensis*) | 0.77 | 45 | 3 | 0.031 | 1.7 |
| Caiman (*Caiman crocodilus*) | 0.78 | 43 | 2 | 0.049 | 0.7 |
| Hippo (*Hippopotamus amphibius*) | 0.76 | 45 | 4 | 0.082 | 2.1 |
| Frog (*Kassina maculata*) | 0.78 | 44 | 1 | | |
| Bearded Dragon (*Pogona vitticeps*) | 0.78 | 48 | 1 | | |
| Crocodile (*Crocodylus palustris*) | 0.78 | 49 | 1 | | |
| Leopard Gecko (*Eublepharis macularius*) | 0.79 | 43 | 2 | 0.034 | 3.5 |
| Tiger Salamander (*Ambystoma tigrinum*) | 0.80 | 40 | 1 | | |
| Monitor Lizard (*Varanus salvator*) | 0.81 | 40 | 1 | | |
| Tortoise (*Centrochelys sulcata*) | 0.82 | 42 | 2 | 0.003 | 2.5 |
| Iguana (*Conolophus pallidus*) | 0.84 | 47 | 1 | | |
| Marine Iguana (*Amblyrhynchus cristatus*) | 0.85 | 41 | 1 | | |
| Mouse* (*Mus musculus*) | 0.86 | 40 | 1 | | |
| Alligator (*Alligator mississippiensis*) | 0.87 | 42 | 3 | 0.046 | 2.7 |
| **Group 2** | | | | | |
| Mongoose (*Mungos mungo*) | 0.60 | 16 | 2 | 0.046 | 2.8 |
| Tapir (*Tapirus indicus*) | 0.61 | 26 | 1 | | |
| Rat (*Rattus norvegicus*) | 0.62 | 36 | 1 | | |
| Brown Bear (*Ursus arctos*) | 0.65 | 17 | 1 | | |
| Hyena (*Crocuta crocuta*) | 0.66 | 10 | 1 | | |
| Cat (*Felis catus*) | 0.66 | 21 | 2 | 0.060 | 3.7 |
| Pig (*Sus domesticus*) | 0.66 | 26 | 1 | | |
| Black Bear (*Ursus americanus*) | 0.67 | 18 | 1 | | |
| Horse (*Equus caballus*) | 0.67 | 24 | 2 | 0.015 | 1.1 |
| Zebra (*Equus quagga*) | 0.67 | 24 | 28** | 0.025 | 2.4 |
| Echidna (*Tachyglossus aculeatus*) | 0.67 | 13 | 1 | | |
| Polar Bear (*Ursus maritimus*) | 0.68 | 16 | 1 | | |
| Rhino (*Ceratotherium simum*) | 0.68 | 21 | 2 | 0.079 | 1.7 |
| Impala (*Aepyceros melampus*) | 0.68 | 19 | 1 | | |
| Lion (*Panthera leo*) | 0.68 | 15 | 3 | 0.017 | 1.5 |
| Sheep (*Ovis aries*) | 0.68 | 28 | 1 | | |
| Giraffe (*Giraffa camelopardalis*) | 0.68 | 14 | 5 | 0.030 | 1.4 |
| Cow (*Bos taurus taurus*) | 0.69 | 29 | 5 | 0.035 | 3.4 |
| Deer (*Odocoileus virginianus*) | 0.69 | 31 | 2 | 0.049 | 9.1 |
| Cheetah (*Acinonyx jubatus*) | 0.70 | 18 | 1 | | |
| Giant Anteater (*Myrmecophaga tridactyla*) | 0.70 | 27 | 1 | | |
| Tapir (*Tapirus terrestris*) | 0.70 | 26 | 1 | | |
| American Buffalo (*Bison bison*) | 0.71 | 16 | 1 | | |
| Bobcat (*Lynx rufus*) | 0.71 | 17 | 1 | | |
| Barbary Sheep (*Ammotragus lervia*) | 0.71 | 26 | 1 | | |

*Table 1 continued on next page*

*Table 1 continued*

| | Median duty factor | Median phase (%) | N | SD duty factor | SD phase (%) |
|---|---|---|---|---|---|
| Raccoon (*Procyon lotor*) | 0.71 | 14 | 1 | | |
| Goat (*Capra aegagrus hircus*) | 0.72 | 23 | 1 | | |
| Llama (*Lama glama*) | 0.72 | 17 | 1 | | |
| Camel (*Camelus bactrianus*) | 0.72 | 21 | 1 | | |
| Wildebeest (*Connochaetes taurinus*) | 0.72 | 18 | 11** | 0.017 | 1.6 |
| Donkey (*Equus africanus asinus*) | 0.73 | 25 | 1 | | |
| Elephant (*Elephas maximus*) | 0.73 | 19 | 1 | | |
| Wombat (*Lasiorhinus krefftii*) | 0.73 | 18 | 1 | | |
| Elephant (*Loxodonta africana*) | 0.74 | 17 | 4 | 0.026 | 1.3 |
| Porcupine (*Erethizon dorsatum*) | 0.75 | 29 | 2 | 0.051 | 0 |

DOI: https://doi.org/10.7554/eLife.29495.003

[*Usherwood, 2016*]) (an argument related to, but distinct from, that proposed in (*Jayes and Alexander, 1980*) for tortoises). The issues of differential scaling between work and power has been covered elsewhere, and related to the scaling of: jumping performance (*Bennet-Clark, 1977*), gait selection and dynamic similarity (*Alexander and Jayes, 1983*), posture (*Usherwood, 2013*) and ontogeny of bipedal gaits (*Hubel and Usherwood, 2015*). It is sufficient here to note that it is not surprising that slow, slow-muscled and small walking quadrupeds operate with high duty factors; one aim of this paper is to account for why phases of 40–50% are adopted in this group.

## Group 2: Duty factors around 0.6 to 0.75

Group 2 covers the majority of self-selected walking observed in upright, medium to large mammals: duty factors between around 0.6 and 0.75, and phases at or somewhat below 25%. A range of exceptions exists, notably some dogs [*Hildebrand, 1968*; *Cartmill et al., 2002*], which may adopt phases close to 0.5 (trotting) at higher speeds (lower duty factors). It appears likely that trotting mechanics can dominate at duty factors close to 0.5, equivalent to grounded running (*Biknevicius and Reilly, 2006*), and we do not attempt to include an account for this within the models of walking presented here.

### Primates

Primates appear to adopt a discretely different limb phasing at the duty factors of Group 2, close to 75% ([*Cartmill et al., 2002*; *Hildebrand, 1967*] sometimes termed a 'diagonal sequence'). Some other quadrupeds (opossum, [*Schmitt and Lemelin, 2002*]) occupy a similar region, potentially due to their grasping forelimbs and/or to cope with the hazards of walking along branches with a risk of failure (*Cartmill et al., 2002*).

### Sloth

Data published for 2-toed sloths locomoting with a suspended, under-branch quadrupedal gait (*Nyakatura et al., 2010*) cover a large range of duty factors. The relationship between duty factor and phase appears to contrast with that found across non-primate quadrupeds (Group 1 vs. Group 2): sloths adopt phases close to 25% at high duty factors (low speed), and 50% at lower duty factors (higher speeds).

### Phases above 0.8

There appears to be no record of a steady gait occurring at this phase in nature.

## Theory methods

### Model development

A numerical model was developed to calculate the mechanical work requirements of all the limbs on the center of mass as a function of duty factor and phase for symmetrical quadrupedal gaits. Mass-normalized vertical, horizontal and lateral forces for each limb were modeled with sinusoidal wave-forms (see *Figure 2*). Vertically, a half-sine force profile $F_{z,\text{limb}}$ was used through time $t$ over the stance period $T_{\text{stance}}$:

$$F_{z,\text{limb}} = A_z \sin(\frac{t}{T_{\text{stance}}}), \tag{1}$$

where amplitude $A_z$ is sufficient to oppose gravity $g$ for a given duty factor DF ($\text{DF} = \frac{T_{\text{stance}}}{T_{\text{stride}}}$, where $T_{\text{stride}}$ is the period of a complete stride cycle):

$$A_z = \frac{g}{8\text{DF}}. \tag{2}$$

Fore-aft forces were modeled with a full sine wave with bias from a half sine-wave:

$$F_{x,\text{limb}} = A_x \sin(\frac{2t}{T_{\text{stance}}}) \pm A_{x,\text{bias}} \sin(\frac{t}{T_{\text{stance}}}), \tag{3}$$

where the amplitude $A_x$, as a proportion of the vertical amplitude, would relate closely to stance angle. The maximum leg angle from vertical $\Phi$ can be approximated by considering an instant a quarter of the way through stance, assuming the combination of $F_{z,\text{limb}}$ and $F_{x,\text{limb}}$ to result in pure compression forces along the leg, and assuming the leg to sweep over the foot at a constant angular rate:

$$\Phi \approx 2\tan^{-1}\left(\frac{A_z\sqrt{2}}{A_x}\right) \tag{4}$$

A single value of $A_z/A_x$ (so a single stance angle) is used; we take no account of any variation in stance angle or stance length with duty factor or speed.

Fore-aft force profiles were supplemented with a bias (shown in the $A_{x,\text{bias}}$ term of *Equation 3*) in which the hindlimbs provide a net forward impulse, counteracted by an equivalent backward impulse from the forelimbs. Finally, small medial forces $F_{y,\text{limb}}$ were introduced, applied equally and medially by each leg. Both of these latter additions follow the half-sine waveform of the vertical force, but with much lower amplitudes (10% and 5% of vertical respectively). At this stage, these values merely serve to demonstrate tendencies; while their influences can easily be modeled, we do not explore this aspect of the parameter space quantitatively here as we have limited data on how they might vary with size, species, speed, duty factor etc.

From the limb force profiles and timings relating to limb phasing, fluctuations in center of mass velocity $\vec{V}_{\text{CoM}}$ were derived by integrating the sum of (mass normalized) limb forces through time (and then mean subtracting to give fluctuations about zero). Assuming a zero mean velocity vertically and laterally, and a positive mean forward velocity over a stride cycle, center of mass velocities were calculated. Note that the value of mean velocity, as long as sufficient to keep instantaneous fore-aft velocities always forward, has no bearing on the model results. The power demanded from each limb is given by:

$$P_{\text{limb}} = \vec{F}_{\text{limb}} \cdot \vec{V}_{\text{CoM}} = F_{z,\text{limb}} V_{z,\text{CoM}} + F_{x,\text{limb}} V_{x,\text{CoM}} + F_{y,\text{limb}} V_{y,\text{CoM}} \tag{5}$$

and the positive components of $P_{\text{limb}}$ – for all legs – provide the mechanical work 'cost'. Note that the instantaneous limb power is the sum of all the components of the right hand side: a positive power due to a vertical force and velocity at the same time as a negative power due to a horizontal force and velocity would cancel; a simple vaulting or pendulum action would be calculated as passive and demand no limb power.

Our approach assumes that the impulses from each limb – and the timing of these forces – act independently from the limb phasing; and that center of mass motions are due to the action of these

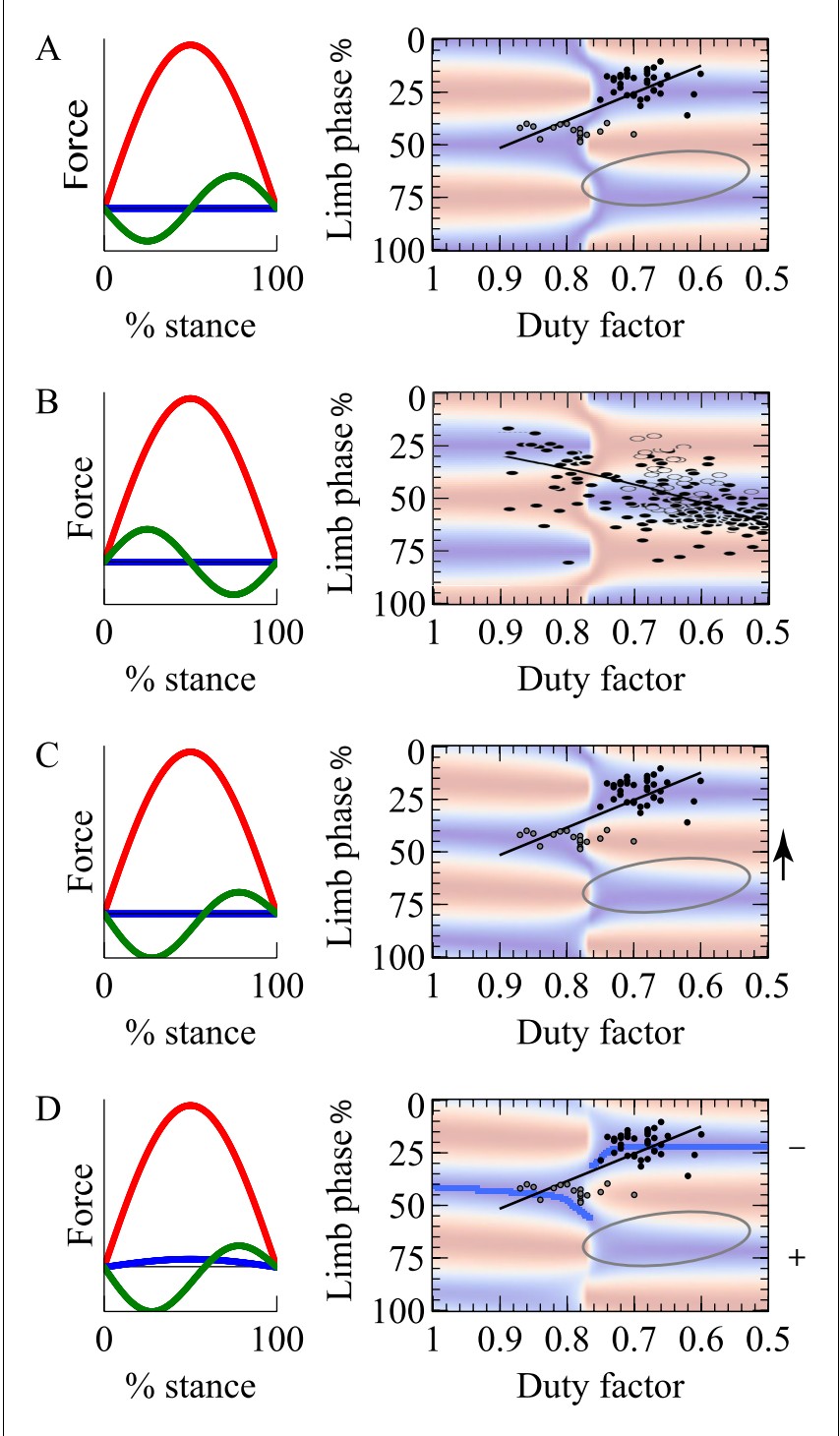

**Figure 2.** Model single limb forces (red vertical, green fore-aft, blue medial) and limb work cost surfaces (red high cost, blue low) for: (**A**) net-vertical limb impulses, and a cyclic decelerate-accelerate fore-aft force; (**B**) net-vertical limb impulses and a reversed fore-aft force profile modeling suspended, sloth-like progression; (**C**) inclined limb impulses, accelerating with the hindlimbs and decelerating with the fore (fore force trace shown); (**D**) as for (**C**) with the addition of a small medial impulse for each limb. Overlying data (**A, C, D**) from a diverse range of quadrupedal species (see *Figure 1*), or only 2-toed sloth (**B**, [*Nyakatura et al., 2010*]). Overlying grey oval (**A,C,D**) denotes the region identified for primates by (*Cartmill et al., 2002*). Model C shifts the cost surface towards lower phases (arrow); Model D tips the surface, making higher phases more costly (denoted by +/−), and consequently the lower phase trough becomes the global minimum for each duty factor (dark blue line).

DOI: https://doi.org/10.7554/eLife.29495.004

limb forces. We assume that 'limb work' is costly – that negative work being performed by one limb cannot power the positive work being performed by another limb at the same time (as would be the assumption if only center of mass or 'external' work was being considered). In other words, simultaneous positive and negative power of equal magnitudes from different limbs would not change the mechanical energy of the centre of mass (no 'external work' would be performed) but would require mechanical power from a limb, and so impose a cost according to the current model. Further, all positive work is considered costly, with no capacity for recovery of negative work through elasticity. While a variety of cost functions may be important (peak power, muscle force etc.) – as discussed in (*Usherwood, 2016*) – we assume here that it is the sum of mechanical limb work – or a direct correlate thereof (which may include peak instantaneous power) – that is relevant to the animal.

While this method has much in common with the pioneering analytical work of Alexander from the 1980's (*Jayes and Alexander, 1980*; *Alexander and Jayes, 1980*; *Alexander, 1989*), advances in computer capability now allow additional parameter spaces – including mediolateral and fore-aft force fluctuations – to be explored numerically without demanding detailed mathematical derivations.

Our approach to studying work-minimizing strategies does *not* directly address the motivations behind the selection of stance angles, duty factors, fore-aft or lateral impulses, or details of force profiles more subtle than the sine-wave approximations use here. Clearly, many of these aspects may be related to work minimization provided the constraints of animal form; however, details may also relate to other geometric constraints - such as maximum allowable pitch or roll deviations as proposed by Jayes and Alexander for tortoises (*Jayes and Alexander, 1980*). Further, it should be emphasized that the approach taken here does *not* include any interplay between limb phasing and impulses or timing of forces through the stance; it is therefore incapable of approaching some aspects of passive vaulting or falling mechanics generally considered important in walking at above moderate speeds. Our approach therefore has limitations: many aspects of both energetics and stability (if indeed these two aspects should be separated) remain to be explored (e.g. [*Wilshin et al., 2017*]).

## To summarize the key caveats

the model makes no attempt to account for the selection of duty factor – it merely calculates the implications of different phases for given duty factors and force profiles. It assumes limb forces through time are not influenced by phasing; while this appears broadly reasonable at moderate speeds, where fluctuations in horizontal velocity are small compared with mean velocity (and so the timecourse of changes in leg angle varies little), it certainly fails at exceedingly low speeds with near-stationary periods (as in grazing). The model is exceedingly simple: no account is taken of lateral bending, belly or tail ground contact, all of which may be relevant to some (but not all) amphibians and reptiles.

This paper may, though, provide a default starting point for considering the mechanical energetic implications of footfall timing in quadrupedal walking, especially at moderate speeds.

## Geometrical accounting for model results

An ideal rolling wheel can support the weight of a vehicle moving over level ground without performing mechanical work because the forces opposing weight are orientated perpendicular to the velocity of the mass. The same is not true when driving up hill: some component of body weight is in the direction of velocity, and an engine is required to deliver power. The value of this principle, that the orientation between the force (or impulse) and velocity accounts for the demand for mechanical power (or work), can be couched in terms of collision mechanics. This reduction is becoming to be appreciated in the study of human and animal mechanics (e.g. *Kuo, 2002*; *Ruina et al., 2005*; *Lee et al., 2011*; see *Bertram, 2016* for an overview). We use the principle here to provide tractable, intuitive accounts for the outputs of the numerical models described above. We link the outputs of the numerical, sine-based models to the principles of collision mechanics by highlighting the angles between center of mass velocities and limb forces at key instants, or the net effect of limb forces in the form of impulses. This approach to explaining the principles underlying differences in mechanical work demand supports the case that the mechanisms found are reasonably general and not merely a mathematical peculiarity of the specific sine-based assumptions of the model.

When approaching work calculations from a collisional perspective, the angle between the limb force (or impulse) and center of mass velocity is critical. It should be highlighted that, with multiple limbs applying forces to the center of mass at the same time, the relationship between limb forces and changes in center of mass velocity is not constant, and may not be intuitive. The geometry provides equal insight in terms of both the mechanical work 'lost' or 'demanded' – all gaits considered here balance the negative and positive works over a cycle. In the discussion, we generally focus on the mechanisms influencing the *loss* of energy; however, the process of returning this energy presumably dominates the physiological cost.

## Model results and discussion

### Numerical model results and mechanistic accounts

The simplest form of the model (*Figure 2A*) has vertical limb forces of a half sine-wave of sufficient amplitude to support body weight (increasing with decreasing duty factor) and horizontal forces of a sine-wave of a suitable amplitude – we choose a value of $-0.2$ the vertical amplitude, approximately equivalent to a stance sweep-angle of 30 degrees. All results are broadly insensitive to this assumption; systematic changes in stance angle with speed or duty factor are not included, but in any case are negligible after the normalization at each duty factor. The implications of duty factor and phase can be calculated in terms of limb work, and presented as a cost surface, normalized for each duty factor (blue minimum, red maximum for each duty factor). The surface shows work minimization at phases of 50% and 0/100% at duty factors above 0.75, and 25%/75% at duty factors below 0.75. The DF = 0.75 cut-off is not exact; it is slightly higher with increasing horizontal force amplitude – or stance angle.

At low duty factors (0.5 < DF < 0.75), the vertical forces produced by the legs relate closely to the vertical forces experienced by the center of mass (see *Figure 3*, which shows a 50% phasing). In this situation, center of mass velocity is forward and downward in the first half of stance (*Figure 3C*; *Figure 4A*), while it is met with an upward and decelerating force, resulting in negative work. The angle between the velocity and force vectors would be improved – made more nearly perpendicular – with a lower magnitude of vertical velocity. This would be achieved by distributing the vertical impulses evenly through time through using a 25% or 75% footfall phasing – benefits analogous to having more spokes on a rimless wheel. In contrast, at duty factors above 0.75, there is sufficient overlap for the forces produced by the limbs to combine such that sub-maximal vertical forces on the center of mass occur when an individual limb is maximally loaded. This results in a reversal in timing of upward and downward velocities compared with lower duty factors (*Figure 3*), at which point higher magnitudes of vertical velocity are favorable, as they increase the angle between center of mass velocity and limb force vectors (*Figure 4A*), reducing the limb power. This principle is analogous to that in bipedal walking with vaulting, stiff-limbed 'inverted pendulum'

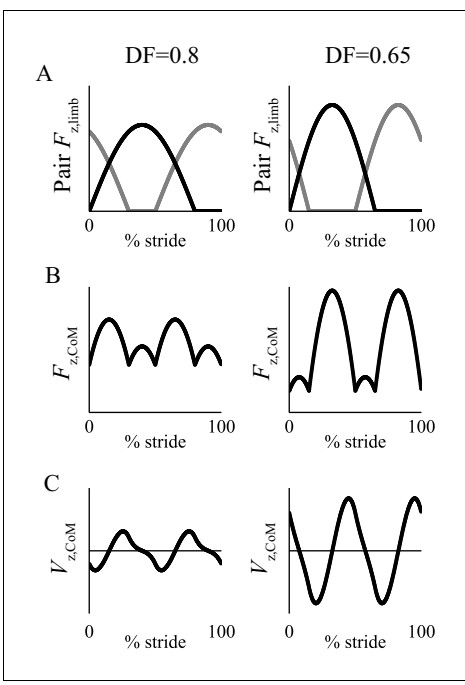

**Figure 3.** Graphical representation of the change of the relationship between vertical limb forces of diagonal pairs for a phase of 50% (trot timing) (A), center of mass forces (B) and center of mass velocities (C) due to high duty factor (0.8, suitable for Group 1) and low (0.65, Group 2). At high duty factors, the vertical limb forces combine such that peak vertical force on the center of mass does not occur at the same time as the peak vertical limb force; consequently, whereas the vertical velocity of the center of mass is predominantly downward during the first half of stance in low duty factors, they are predominantly upward over the same period at high duty factors.
DOI: https://doi.org/10.7554/eLife.29495.005

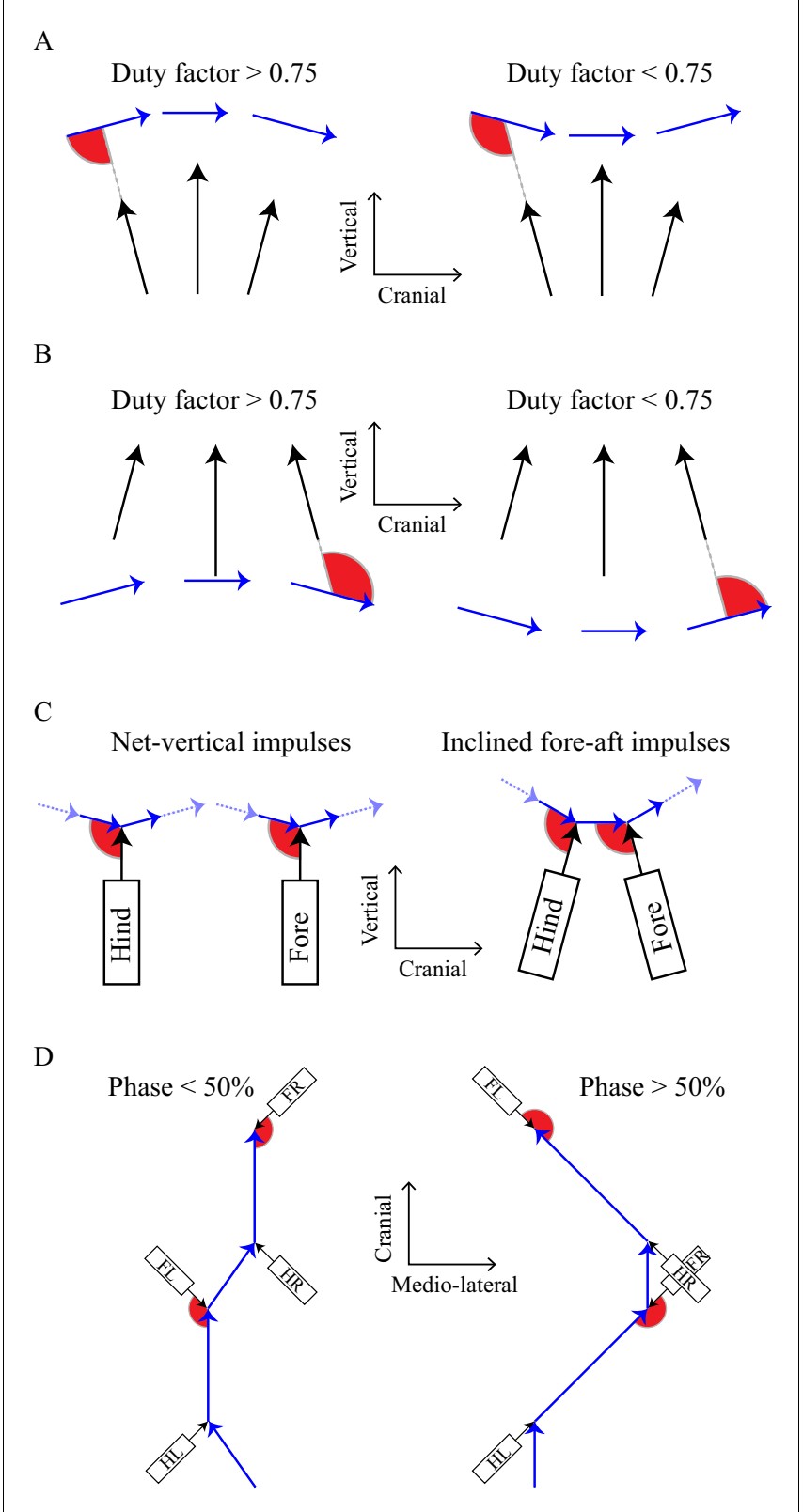

**Figure 4.** Geometric accounts for the cost surfaces found with the numerical model, highlighting the angle between limb forces or impulses (black arrows) and center of mass velocities. Losses are lower if the angles (red wedges) between limb force (or impulse) and center of mass velocity vectors become closer to perpendicular. A): low mechanical work is demanded at 0/50% at high duty factors and 25/75% with low duty factors. B): the reversed

*Figure 4 continued on next page*

*Figure 4 continued*

relationship for sloths, in which horizontal forces profiles are reversed (see *Figure 2B*). In (A) and (B), force and velocity vectors are shown at snapshots in early, mid and late stance. 0% and 50% phasing (pace/trot timing) results in higher magnitude of center of mass vertical velocity, making the angle between limb force and center of mass velocity more nearly perpendicular at high duty factor (A), but at low duty factors for suspended, sloth-like gaits (B). C: the effect of inclined fore-aft impulses, demonstrated for low duty factors. Losses for purely vertical net impulses are minimized with even timing because changes in velocity vector are divided evenly; when the hind limbs impose net-acceleratory impulses (resisted by deceleration from the forelimbs), an even division of these collisions requires uneven phasing, reducing the period behind hind and fore contact (reducing the work-minimizing phase). D: a geometric account for the cost of high phases. Limb impulses (for left hind, LH, left fore, LF, right hind RH and right fore RF) and center of mass velocities are shown in the x-y plane (mediolateral/fore aft) for low and high phases. Again, angles between limb impulses and center of mass velocities are more favorable – closer to perpendicular – in the phase predicted to have lower cost by the numerical model. This is also consistent with lower fluctuations in speed (arrow length) – and so kinetic energy – for equivalent mean forward velocities.
DOI: https://doi.org/10.7554/eLife.29495.006

mechanics. The beneficial, higher vertical velocity magnitudes at the beginning of stance are achieved with synchronous footfalls – phases of 0% or 50% (pace or trot).

Suspended quadrupedal walking, as typical for sloths and used facultatively in a range of primates, provides a useful contrast. In this case, the sense of the horizontal forces is reversed (*Figure 2B*). In early stance each limb – acting predominantly in tension instead of compression – provides an accelerating force, and decelerates over the second half of stance (confirmed by force measurements in lemurs, [*Granatosky et al., 2016*]). With this difference, force-velocity angles are improved (*Figure 4B*) and work minimization is achieved with a reversed duty factor/phase relationship: high duty factor (slow) gaits benefit from low vertical velocity magnitudes and distributed foot timing; low duty factor (faster) gaits benefit from higher vertical velocity magnitudes and synchronous (trotting or pacing) foot contact phasing. This relationship appears to be supported by observations of locomoting sloths (*Nyakatura et al., 2010*).

The extension of the model to include net foreword impulses from the hind limbs and rear-ward impulses from the forelimbs (*Figure 2C*) is achieved with a fore-aft bias of an additional half sine-wave (positive for hindlimbs; negative for fore) of an amplitude 0.1 that of the vertical. The cost surface is shifted towards lower phases, with low-cost regions becoming <50% (100%) at high duty factor and <25% (75%) at low duty factors. The extent of this shifting increases with higher bias amplitudes.

The mechanism underlying this shift can again be described using the collisional framework (*Figure 4C*). In this case, the net impulse from each limb is treated as a single vector, orientated slightly forward for hindlimbs and backward in forelimbs. With this inclination the angles between velocities and impulses are most favorable – most near perpendicular – with a reduced interval between hind and forelimb contacts. This is the principle described by Ruina *et al.* (*Ruina et al., 2005*) to account for the 'gathered' gallop of horses, and is related to a previously speculated mechanism for walking dogs (*Usherwood et al., 2007*).

Up to this stage of model development, everything could be treated in a planar manner: there is no difference between left and right, and phases ± 50% are equivalent. The final addition to the model is small medial impulses (as generally reported, when measured ((e.g. dogs: [*Griffin et al., 2004*]) and considered of relevance) produced by a half sine-wave of amplitude 0.05 that of vertical (*Figure 2D*). The effect of this addition on the cost surface it to tip the low phases to be less costly; and make phases > 80% uneconomical. The cost at the troughs (blue regions) for each duty factor become slightly different; the dark blue line in *Figure 2D* denotes the global limb work minimum for each duty factor.

The mechanism underlying this change can again be related to collision mechanics (*Figure 4D*), this time considering only the horizontal plane. Each hindlimb produces an impulse that accelerates the body both forward and medially; each forelimb provides backward and medial impulses. *Figure 4D* demonstrates the difference between low and high phases: with low phasing, each impulse acts more closely to perpendicular to the center of mass velocities, and so lower mechanical limb work is required. With the addition of medial impulses, the undesirability of a 95% phase

becomes apparent. Just prior to front-right foot contact, the center of mass velocity is a near-maximal downward, forward and lateral, before being opposed by an impulse orientated upward, backward and medially: much of the kinetic energy of the body cannot be maintained with small-angle collisions. This phenomenon, and the importance generally of timing on the energetic consequences of impulses, can be experienced directly. A human can be asked to crawl with a pacing footfall pattern – this is usually achieved with ease. A phasing just off pacing, with the knee landing just before the hand, is also easily achieved with comfort (phase roughly 5%). However, the slight change the other way, with the right hand landing just prior to the right knee (phase around 95%) is physically very demanding, and very challenging to maintain for any duration.

## Further discussion

Both the limb-force-driven model and the geometric description of mechanism provide energetic accounts for selection of limb phasing in many quadrupeds undertaking slow locomotion. Phases somewhat below 50% are predicted at $DF \approx 0.8$; somewhat below 25% at $DF \approx 0.65$, with the reverse relation predicted for suspended sloth-like quadrupedal gaits. This provides a simple work-minimizing account for commonalities in quadrupedal gaits (*Figure 2C*) across extreme phylogenetic distance (Group 1 – *Figure 1* – includes species that last shared a common ancestor around 300 million years ago), considerable difference in scale (Group 1 covers from hippo to mouse – a 500,000-fold difference in mass) and contrast in muscle properties (very 'slow' (tortoise [*Woledge, 1968*]) to 'fast' (mouse, e.g. [*Askew and Marsh, 1997*])). Group 2 (*Figure 1*), consisting of both familiar and exotic medium to large mammals walking at lower duty factors, approaches the even footfall or slightly lower phases predicted from simple planar collisional principles (*Ruina et al., 2005*; *Usherwood et al., 2007*). The advantage in terms of work minimization of near-25% vs. near-75% phase only becomes apparent when medial impulses are included (*Figure 2D*; *Figure 4D*).

A purely energetic account is not found for the unusual phases used by many primates (and opossum, [*Schmitt and Lemelin, 2002*]); however, the cost landscape is such that the phases observed, while not meeting the absolute energetic minimum, may be in a region that achieves a nearly equivalent performance (see also [*Sellers et al., 2013*]). Some other account (perhaps associated with managing locomotion on unreliable substrates – [*Cartmill et al., 2002*]) is presumably required but beyond the scope of this study. Further support for the explanatory power of the model comes from cases where duty factors transitions over 0.75. The sloth data set has already been discussed. The case of a macaque through ontogeny ([*Hildebrand, 1967*]; a generally consistent theme appears to hold across a range of primates, [*Hurov, 1982*]) also appears to fit model predictions very nicely (*Figure 5*). When very young, the macaque used a very high duty factor, and phase just below trotting. As it matured, it adopted lower duty factors and both horse-like and primate-like phases, but avoided the previous phase. As adult, it adopted a purely adult primate-like phase. As discussed above, the models presented here cannot account for the adult primate phase, but can account for the avoidance of phases around 40% at low duty factors.

The driver behind the high duty factors of 'reptile', slow, small or juvenile gaits, while potentially related to the avoidance of excessive muscle activation for power demands, is not covered here. However, *given* high duty factors, phasing around 0.4 to 0.5 is consistent with limb work minimization. This provides an alternative account for the observation of these near-trotting (i.e. 0.4 to 0.5) phases in human infants (*Patrick et al., 2009*; *Patrick et al., 2012*): where previously they have been attributed to some immaturity in neural development

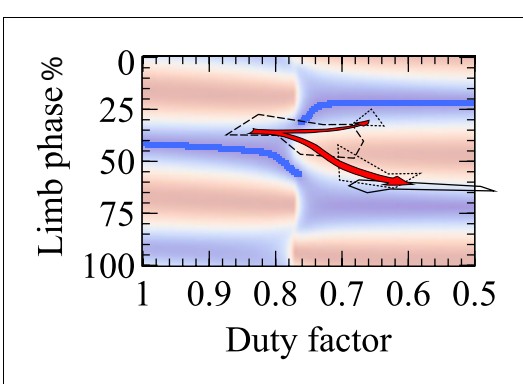

**Figure 5.** The final cost surface (blue low, red high, blue line indicating global minimum for each duty factor) overlain with Hildebrand's measurements (*Hildebrand, 1967*) for a macaque (*Macaca mulatta*) through development. Between 18 and 42 days (dashed line), high duty factors and phases close to 40% are observed. At 52 to 96 days (dotted lines), two regions are observed, before reaching the adult condition (solid line) typical of primates.
DOI: https://doi.org/10.7554/eLife.29495.007

(*Patrick et al., 2012*), they may actually be adaptive from a work minimization perspective. This has clear parallels with the contrasts between toddler and adult walking and running gaits (*Hubel and Usherwood, 2015*): might it be that we would do better viewing ontogeny of gait – whether quadrupedal or bipedal – *first* from an adaptive energetic framework before invoking some constraining limitation in the rate of neural development?

## Acknowledgements

The support and contributions of many members of the Structure and Motion Lab, past and present, are greatly appreciated; in particular, Tatjana Hubel, Vivian Allen, Sandy Kawano, and Andrew Spence (see [Spence et al., 2013]) for sharing their videos of a range of species (see Supplementary ile 1).

## Additional information

### Funding

| Funder | Grant reference number | Author |
|---|---|---|
| Wellcome | 202854/Z/16/Z | James R Usherwood<br>Zoe T Self Davies |

The funders had no role in study design, data collection and interpretation, or the decision to submit the work for publication.

### Author contributions

James R Usherwood, Conceptualization, Resources, Formal analysis, Supervision, Funding acquisition, Visualization, Methodology, Writing—original draft, Project administration, Writing—review and editing; Zoe T Self Davies, Data curation, Investigation, Methodology, Project administration

### Author ORCIDs

James R Usherwood (iD) http://orcid.org/0000-0001-8794-4677
Zoe T Self Davies (iD) http://orcid.org/0000-0003-1419-8735

### Decision letter and Author response

Decision letter https://doi.org/10.7554/eLife.29495.010
Author response https://doi.org/10.7554/eLife.29495.011

## Additional files

### Supplementary files

• Supplementary file 1. All duty factor and phase data for walking quadrupeds, and their sources.
DOI: https://doi.org/10.7554/eLife.29495.008

• Transparent reporting form
DOI: https://doi.org/10.7554/eLife.29495.009

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
