## [Decision Letter]

[Editors’ note: a previous version of this study was rejected after peer review, but the authors submitted for reconsideration. The first decision letter after peer review is shown below.]

Thank you for choosing to send your work, "Work minimization accounts for footfall phasing in slow quadrupedal gaits", for consideration at *eLife*. Your submission has been reviewed by a Senior editor and three reviewers, one of whom is a member of our Board of Reviewing Editors. Although the work is of interest, we regret to inform you that the findings at this stage are too preliminary for further consideration at *eLife*.

Specifically, whereas we found considerable merit in the manuscript, there are a number of substantive issues to be addressed here and one in particular, namely, the use of n=1 for each species, the lack of a clear description of how these samples were selected from YouTube, and, finally, the full reliance on YouTube videos to support the main scientific point.

The editors of the journal and reviewers concur that YouTube based data is insufficient to provide the rigorous support required here. We do think that YouTube data can support the broader relevance of the main point in a paper, but the core evidence needs to be corroborated from rigorous lab or field data collected by researchers.

Whereas we are rejecting the manuscript in its current form, we believe that it should be entirely possible to include rigorous lab and field measurement data based on RVC's terrestrial locomotion research program. Further, an additional significant increase of the YouTube-based sample size using a well-established (lab / field data-informed) and well-described screening process is needed. We do appreciate how these YouTube videos enable the sampling of species beyond the RVC terrestrial locomotion research program, combining both has the potential to support an especially compelling synthesis.

Considering the required effort to meet *eLife* expectations is deemed more than two months of work we are rejecting the paper at this stage, given *eLife*'s policy to only ask for revisions when the work can be completed in two months. However, we expect that you should be able in principle to collect these data drawing on the archives at RVC, recording locomotion at a local zoo, and by expanding your YouTube search relying on more rigorous selection processes. We do appreciate how the models and scientific ideas in the paper can advance our understanding of the economy of terrestrial locomotion, they would be suitable for re-evaluation when corroborated from a more complete set of scientific observations.

*Reviewer #1:*

The strengths of this paper include that the modeling, the general framework on the manuscript, has a lot of merit and is of broader interest. The part that I found disappointing includes that the YouTube data is something that seems more appropriate for the last broader impact figure of a manuscript. This could function to illustrate how more rigorous lab or field results analyzed and discussed earlier can be extrapolated to more species, however the YouTube data is all the data presented so this manuscript is rather thin. This is N=1, n=1 data per species for about 40 species studied based on YouTube videos, although the number of animals per species and trials are not explicitly written, and the similarity between trials are unclear too, the std and other information are missing based on which I inferred this conclusion.

It seems like many more YouTube videos could be found for several of the species to better support the claims with statistics (N=3 individuals). A visit to the zoo could be sufficient for recording walking phases too. Much of the data is rather preliminary, and one would hope to see actual force plate and motion capture data from lab studies to backup model assumptions for a few species to demonstrate the effect more thoroughly (the world-class infrastructure of the group of the author enables such measurements). Perhaps more practically; the lab of the author already has a lot of data on many animals species archived based on many years of impactful research of the RVC Structure and Motion lab, this wealth of data could be better sampled and integrated to address the comment.

Having said this, a lot of data is not essential for a good paper, but I think it is essential to back up the claims made for a journal like *eLife*, especially due to strong claims such as "Phases predicted to be particularly costly are not observed in nature", which are awkward considering the small number of videos studied, and the fact that many of these animals actually locomote in complex terrains that make up their habitats, which is understudied and not discussed. Hence, I have reasons to believe that some of the claims in this manuscript are not as rigorously supported as we would like to see; the study is promising though.

The issues can be addressed by including data from the RVC archives, more YouTube videos and as needed some videos made in the local zoo. It would also help to remove some of the strong wording in the manuscript.

Finally, the manuscript is not that well written, the short Abstract upfront is a hard read, the other Abstract and significant statement are good. In the introduction there are some exceptionally long and poorly structured sentences that are unreadable without staring and going back and forth. In general the manuscript needs a good polish.

In conclusion I am encouraging the authors to polish the manuscript, tone it down and integrate the RVC terrestrial locomotion research data from the past many years as well as more YouTube videos per species so proper statistics can be conducted per species and for the total dataset. With these changes it would be of interest for publication in *eLife*.

*Reviewer #2:*

This manuscript uses a model and gait data collected largely from YouTube to test a model of work minimization in gait. The model is used to explain an observed pattern of clustering of combinations of limb phase and duty factor in different groups of animals. The model shows regions of low work demands that are explained by different collisional costs of different combinations of gait patterns.

A strength of the study is that the model is very simple. Of course with simplicity comes simplifying assumptions, and I had trouble convincing myself (and didn't find that the authors convinced me) that some of the ignored biological reality might be critical to the interpretation and validity of the model. A few particular known phenomena seem particularly important and at least should be acknowledged, if not dealt with. First, most of the group one animals have a gait that includes significant lateral undulation. None of the group two animals do. This might be an influence on phasing, and it might also influence the pattern of ground reaction forces that are modeled. Group 1 animals also adopt a sprawled limb posture, which would seem very relevant to the model assumption and designs described in subsection “Model development”. Some of the group 1 animals "walk" with their bellies dragging on the ground. Also, small animals tend to brake with their forelimbs and accelerate with their hindlimbs, whereas large animals distribute both tasks across all limbs. The modeling does not seem to account for this, or address whether it might influence the results. It seems to me incorporating such a known pattern might significantly alter the results.

A simplification that I also found troubling is what seems to be a model of the quadrupedal gait as bipedal (Figure 3). I liked this figure and it helped me understand the model outputs. I could imagine how the simplification of focusing on just two limbs might be justified (although I didn't see a clear justification), but I could also imagine that a bipedal representation may not be appropriate. Limb cycles in quadrupedal walking are complex, and vary with phasing. The left and right sides are doing different things at different times. It would seem to me that a comparison of c.o.m. forces and limb forces would have to account for the phasing relationship between all four limbs.

As the application and interpretation of the model is so divorced from biological reality, it limits what it can tell us. It tells us perhaps that collisional mechanics are important and might shape footfall patterns. That by itself doesn't seem very surprising. A discussion of the biological realities mentioned above might serve not only to address critiques of the model, but to further inform our understanding of variation in how different animals move.

My concerns might be addressed by a presentation that does more to apply more skepticism to the model, and to connect it more solidly to the biology in its interpretation.

*Reviewer #3:*

The manuscript focuses on the effect of fore-hind leg phasing on external mechanical work during walking. The authors compute theoretical cost surfaces in duty factor and fore-hind timing space. The results support that collision reduction strategies are used generally in quadrupedal walking, similar to the consensus on bipedal walking and overall running mechanics.

The manuscript is captively written using examples that will have readers personally test work minimization. Most assumptions of the modeling efforts are accurately addressed. I expect the content to be of considerable interest to a broad audience.

I only have a few comments and recommend acceptance with minor revisions.

Comments:

State and justify the underlying assumption that muscle work is proportional to metabolic cost. A major underlying assumption of the collision reduction paradigm is that mechanical work requires proportional muscle work, and that increased muscle work incurs an increased metabolic cost. Force production with muscle is not free. Moreover, the cost of muscle producing isometric force is not necessarily lower than the cost of muscle producing force while going through work cycles (see Holt et al., JEB 2014). Please address these assumptions.

Presumably, Vcom followed from integration of the limb force profiles. In order to obtain exact solutions of this integration, and thus CoM trajectories, an initial Vcom would be required. Please specify if CoM trajectories were determined, and if so, how the initial Vcom was determined. Different initial Vcom values may result in different CoM trajectories and thus cost surfaces, no?

Please use proper vector notation in Equation 5. As the authors point out, and perpendicular application of force to the CoM vector would result in no work. This is accurately reflected by the dot product between Flimb and Vcom. However, the assertion that "the positive components of which… provide the mechanical work 'cost' " is unclear. Please clarify if any positive components of the RHS of Equation 5 are taken as 'cost', or whether only positive parts of the LHS, Plimb, are considered 'cost'. To infer collision reduction strategies, the latter seems appropriate, and a clarification would be helpful.

---

## [Author Response]

[Editors’ note: the author responses to the first round of peer review follow.]

Specifically, whereas we found considerable merit in the manuscript, there are a number of substantive issues to be addressed here and one in particular, namely, the use of n=1 for each species, the lack of a clear description of how these samples were selected from YouTube, and, finally, the full reliance on YouTube videos to support the main scientific point.The editors of the journal and reviewers concur that YouTube based data is insufficient to provide the rigorous support required here. We do think that YouTube data can support the broader relevance of the main point in a paper, but the core evidence needs to be corroborated from rigorous lab or field data collected by researchers.Whereas we are rejecting the manuscript in its current form, we believe that it should be entirely possible to include rigorous lab and field measurement data based on RVC's terrestrial locomotion research program. Further, an additional significant increase of the YouTube-based sample size using a well-established (lab / field data-informed) and well-described screening process is needed. We do appreciate how these YouTube videos enable the sampling of species beyond the RVC terrestrial locomotion research program, combining both has the potential to support an especially compelling synthesis.Considering the required effort to meet eLife expectations is deemed more than two months of work we are rejecting the paper at this stage, given eLife's policy to only ask for revisions when the work can be completed in two months. However, we expect that you should be able in principle to collect these data drawing on the archives at RVC, recording locomotion at a local zoo, and by expanding your YouTube search relying on more rigorous selection processes. We do appreciate how the models and scientific ideas in the paper can advance our understanding of the economy of terrestrial locomotion, they would be suitable for re-evaluation when corroborated from a more complete set of scientific observations.

We greatly appreciate the time and attention invested by the editors and reviewers in considering the manuscript. We are pleased that the models and concepts are viewed as presenting a worthwhile scientific advance concerning our understanding of the economy of terrestrial locomotion.

The overriding concern with the initial submission appears to be with the use of YouTube video to provide data. It is not clear what the specific issue is, so I should perhaps clarify that: absolute timings, angles, distances or speeds are not used in calculating either Duty Factor or Phase, so there is no requirement for a known frame rate, perpendicular view nor calibrated field. It is therefore difficult to appreciate why the recording of video should be limited to ‘researchers’.

What is more, any concern that the animals might not be meeting the selection criteria – of approximately steady walking over flat, level ground – can immediately be assessed from the YouTube links – something that is rarely possible with data from ‘researchers’; even with the advent of electronic supplementary information, it is exceptionally rare for all the source video to be made freely and immediately available.

A somewhat separate concern is that our original sampling of 38 YouTube observations from 38 species might be insufficient. To address this, we have increased our sampling to 120 observations of 52 species, including multiple samples (notably zebra and wildebeest) from ongoing projects at the Structure and Motion lab. The effect of including these data has been minimal: the gradient of the regression line relating phase to duty factor changes from

Phase (%)= 128DF – 66

To

Phase (%) = 130DF – 66

and the informal grouping used in the original manuscript is consistent with a Kmeans clustering analysis now included.

While many species now have multiple measurements, we continue to include those with single observations. If we can assume standard deviations to be consistent between species, the single measurements are likely (95%) to be within 0.08 of the mean DF and within 5.2% of the mean phase.

We therefore feel secure in our assertion that there is a relationship between duty factor and phase, and that reptiles and amphibians tend to use a high duty factor and phase around 45%, while most moderate to large mammals use a lower walking duty factor and phases closer to 25%.

We should continue to note that we do not claim novelty in these observations. Duty factor and phase of walking quadrupeds have been reported extensively with ‘*rigorous lab or field data collected by researchers*’ cited in Supplementary file 1. The contribution of Hildebrand is especially highlighted. We note relevant values from previous published work, but do not feel it appropriate to replicate this. For instance, elephants have been measured by the Structure and Motion Lab (Hutchinson et al., 2006): 42 Asian and 14 African elephants over 2400 strides. Ongoing work at the lab will publish further gait data on different species, but it is inappropriate to publish these first here. We are therefore limited to observations from the lab that have either been published (such as the elephants) or are incidental to the relevant project (such as the zebra and wildebeest walking in front of automatically triggered cameras in Botswana).

In view of the broadly positive initial reviews, reassurances concerning the appropriateness of YouTube in providing source video, and the expanded sampling of both YouTube and locally sourced video, we hope that you reconsider this paper.

Reviewer #1:The strengths of this paper include that the modeling, the general framework on the manuscript, has a lot of merit and is of broader interest. The part that I found disappointing includes that the YouTube data is something that seems more appropriate for the last broader impact figure of a manuscript. This could function to illustrate how more rigorous lab or field results analyzed and discussed earlier can be extrapolated to more species, however the YouTube data is all the data presented so this manuscript is rather thin. This is N=1, n=1 data per species for about 40 species studied based on YouTube videos, although the number of animals per species and trials are not explicitly written, and the similarity between trials are unclear too, the std and other information are missing based on which I inferred this conclusion.It seems like many more YouTube videos could be found for several of the species to better support the claims with statistics (N=3 individuals). A visit to the zoo could be sufficient for recording walking phases too. Much of the data is rather preliminary, and one would hope to see actual force plate and motion capture data from lab studies to backup model assumptions for a few species to demonstrate the effect more thoroughly (the world-class infrastructure of the group of the author enables such measurements). Perhaps more practically; the lab of the author already has a lot of data on many animals species archived based on many years of impactful research of the RVC Structure and Motion lab, this wealth of data could be better sampled and integrated to address the comment.

These issues are discussed above.

Having said this, a lot of data is not essential for a good paper, but I think it is essential to back up the claims made for a journal like eLife, especially due to strong claims such as "Phases predicted to be particularly costly are not observed in nature", which are awkward considering the small number of videos studied, and the fact that many of these animals actually locomote in complex terrains that make up their habitats, which is understudied and not discussed. Hence, I have reasons to believe that some of the claims in this manuscript are not as rigorously supported as we would like to see; the study is promising though.

This ‘not observed’ claim is not really based on the measurements reported here, but on the thirty or so years of work from Hildebrand.

The issues can be addressed by including data from the RVC archives, more YouTube videos and as needed some videos made in the local zoo.

We have increased our sampling from 38 YouTube observations of 38 species to 120 observations of 52 species, including multiple samples (notably zebra and wildebeest) from ongoing projects at the Structure and Motion lab. See Response to Senior Editor for expanded discussion.

It would also help to remove some of the strong wording in the manuscript.

I agree that ‘not observed’ may be viewed as an inference too far (but an inference based on an awful lot of published work), so the Abstract is adjusted to:

“Phases predicted to be particularly costly have not been reported in nature.”

We assume that the reader will understand ‘during steady walking on level ground’ by context.

Finally, the manuscript is not that well written, the short Abstract upfront is a hard read, the other Abstract and significant statement are good. In the Introduction there are some exceptionally long and poorly structured sentences that are unreadable without staring and going back and forth. In general the manuscript needs a good polish.

It is difficult to resolve this without specifics and without creating a conflict with reviewer 3 comments (I presume ‘captively written’ is a good thing…?). However, hopefully some of our revisions will clarify things.

In conclusion I am encouraging the authors to polish the manuscript, tone it down and integrate the RVC terrestrial locomotion research data from the past many years as well as more YouTube videos per species so proper statistics can be conducted per species and for the total dataset. With these changes it would be of interest for publication in eLife.

A ‘key caveats’ summary section should have helped ‘toning down’ the manuscript.

*Reviewer #2:*

This manuscript uses a model and gait data collected largely from YouTube to test a model of work minimization in gait. The model is used to explain an observed pattern of clustering of combinations of limb phase and duty factor in different groups of animals. The model shows regions of low work demands that are explained by different collisional costs of different combinations of gait patterns.A strength of the study is that the model is very simple. Of course with simplicity comes simplifying assumptions, and I had trouble convincing myself (and didn't find that the authors convinced me) that some of the ignored biological reality might be critical to the interpretation and validity of the model.

A summary of key caveats is now included:

“To summarize the key caveats:

The model makes no attempt to account for the selection of duty factor – it merely calculates the implications of different phases for given duty factors and force profiles;

It assumes limb forces through time are not influenced by phasing. While this appears broadly reasonable at moderate speeds, where fluctuations in horizontal velocity are small compared with mean velocity (and so the timecourse of changes in leg angle varies little), it certainly fails at exceedingly low speeds with near-stationary periods (as in grazing);

The model is exceedingly simple: no account is taken of lateral bending, belly or tail ground contact, all of which may be relevant to some (but not all) amphibians and reptiles.”

A few particular known phenomena seem particularly important and at least should be acknowledged, if not dealt with. First, most of the group one animals have a gait that includes significant lateral undulation.

And yet, many also do not: Hippo? Tortoise? Lateral bending may make the model less accurate (noted in caveats), but it would appear inappropriate to add another level of model complexity before being certain the simplest model is inadequate.

None of the group two animals do. This might be an influence on phasing, and it might also influence the pattern of ground reaction forces that are modeled. Group 1 animals also adopt a sprawled limb posture, which would seem very relevant to the model assumption and designs described in subsection “Model development”. Some of the group 1 animals "walk" with their bellies dragging on the ground.

As above.

Also, small animals tend to brake with their forelimbs and accelerate with their hindlimbs, whereas large animals distribute both tasks across all limbs. The modeling does not seem to account for this, or address whether it might influence the results. It seems to me incorporating such a known pattern might significantly alter the results.

It appears that even large animals net-brake with their forelimbs during steady walking (horses certainly do). The extent of net braking does vary between animals, sizes and speeds. The modeled effect of net braking (Figure 2) is to shift the surface towards lower phases; a higher braking shifts the surface (and optimal phase) further. We therefore limit our interpretation of the results to predicting work minimization at somewhat less than 50% phase (high DF) or somewhat less than 25% phase (low DF). More quantitative predictions would indeed require further empirical input concerning the net braking impulses of each species.

A simplification that I also found troubling is what seems to be a model of the quadrupedal gait as bipedal (Figure 3). I liked this figure and it helped me understand the model outputs. I could imagine how the simplification of focusing on just two limbs might be justified (although I didn't see a clear justification), but I could also imagine that a bipedal representation may not be appropriate. Limb cycles in quadrupedal walking are complex, and vary with phasing. The left and right sides are doing different things at different times. It would seem to me that a comparison of c.o.m. forces and limb forces would have to account for the phasing relationship between all four limbs.

We apologize for the misleading presentation of this in the original manuscript. The graphs are not meant to indicate a bipedal gait, but a trotting or pacing timing (so synchronous fore/aft forces). This is sufficient to explain the difference in force and vertical velocity phasing as a function of duty factor – and so why moving from a trotting footfall timing to an even (25/75%) timing is needed to minimize mechanical work as the duty factor falls below 0.75. We highlight this with an adjusted figure (axis legend), text and legend.

“Graphical representation of the change of the relationship between vertical limb forces of diagonal pairs for a phase of 50% (trot timing) (A), center of mass forces (B) and center of mass velocities (C) due to high duty factor (0.8, suitable for Group 1) and low (0.65, Group 2).”

The relationships between limb and CoM forces, phasing and duty factor and limb work are indeed complex. This accounts for why the collisional account was only developed after the sine-based modeling.

As the application and interpretation of the model is so divorced from biological reality, it limits what it can tell us. It tells us perhaps that collisional mechanics are important and might shape footfall patterns. That by itself doesn't seem very surprising. A discussion of the biological realities mentioned above might serve not only to address critiques of the model, but to further inform our understanding of variation in how different animals move.

That an explicit work-minimizing reduction might account for the phasing of limbs in walking, and that the principles leading to the variation in work requirement can be (in retrospect) understood from collision mechanics, does appear – if not surprising – at least novel.

My concerns might be addressed by a presentation that does more to apply more skepticism to the model, and to connect it more solidly to the biology in its interpretation.

The ‘key caveats’ section should make (the very reasonable) skepticism about the model more explicit.

*Reviewer #3:*

The manuscript focuses on the effect of fore-hind leg phasing on external mechanical work during walking.

Just a quick clarification on the terminology. This manuscript approaches the sum of the mechanical work of the limbs; some conventions would have the ‘external’ work being only that related to changes in energy of the centre of mass (not the approach here).

The authors compute theoretical cost surfaces in duty factor and fore-hind timing space. The results support that collision reduction strategies are used generally in quadrupedal walking, similar to the consensus on bipedal walking and overall running mechanics.The manuscript is captively written using examples that will have readers personally test work minimization. Most assumptions of the modeling efforts are accurately addressed. I expect the content to be of considerable interest to a broad audience.I only have a few comments and recommend acceptance with minor revisions.Comments:State and justify the underlying assumption that muscle work is proportional to metabolic cost. A major underlying assumption of the collision reduction paradigm is that mechanical work requires proportional muscle work, and that increased muscle work incurs an increased metabolic cost. Force production with muscle is not free. Moreover, the cost of muscle producing isometric force is not necessarily lower than the cost of muscle producing force while going through work cycles (see Holt et al., JEB 2014). Please address these assumptions.

I am afraid that my response to this point must be incomplete at this stage. Issues surrounding the relevant costs are clearly important, but are addressed more fully elsewhere (I include a self-citation). I would argue that for a habitual, costly gait such as walking, evolution of form and posture ‘should’ mean that muscle force would not impose a cost per se – as the muscle ‘should’ only be exposed to the forces required for the demands of work and power production (else why not change gearing?). This is an area that continues to develop and cannot be addressed sufficiently here. Instead, I am explicit that the model assumes that it is work (or a direct correlate thereof) that presents the cost relevant to the animal.

“We assume that ‘limb work’ is costly – that negative work being performed by one limb cannot power the positive work being performed by another limb at the same time (as would be the assumption if only center of mass work was being considered). Further, all positive work is considered costly, with no capacity for recovery of negative work through elasticity. While a variety of cost functions may be important (peak power, muscle force etc.) – as discussed in (31) – we assume here that it is sum of mechanical limb work – or a direct correlate thereof (which may include peak instantaneous power) – that is relevant to the animal.”

Presumably, Vcom followed from integration of the limb force profiles.

Indeed.

In order to obtain exact solutions of this integration, and thus CoM trajectories, an initial Vcom would be required. Please specify if CoM trajectories were determined, and if so, how the initial Vcom was determined. Different initial Vcom values may result in different CoM trajectories and thus cost surfaces, no?

Happily, no. By mean-subtracting after integration, fluctuations about zero are found. When added to mean values (zero vertically or laterally; something positive forward), centre of mass velocities can be found without starting with initial velocities for integration constants (though, of course, these could be derived). This process appears widespread now in calculating CoM velocities and positions from forceplate measurements; mean subtraction and the assumption of a complete cycle does ensure the final height matches the initial height, and so integration drift (visible in the early analyses – Cavagna etc.) is hidden.

The section is expanded to clarify this process:

“From the limb force profiles and timings relating to limb phasing, fluctuations in center of mass velocity VCoM→ were derived by integrating the sum of (mass normalized) limb forces through time (and then mean subtracting to give fluctuations about zero). Assuming a zero mean velocity vertically and laterally, and a positive mean forward velocity over a stride cycle, center of mass velocities were found.”

Please use proper vector notation in Equation 5. As the authors point out, and perpendicular application of force to the CoM vector would result in no work. This is accurately reflected by the dot product between Flimb and Vcom. However, the assertion that "the positive components of which… provide the mechanical work 'cost' " is unclear. Please clarify if any positive components of the RHS of Equation 5 are taken as 'cost', or whether only positive parts of the LHS, Plimb, are considered 'cost'. To infer collision reduction strategies, the latter seems appropriate, and a clarification would be helpful.

Sorry for confusion – the reviewer is obviously correct in his/her interpretation. Hopefully this expanded version makes things more explicit.

“The power demanded from each limb is given by:

Plimb=Flimb→⋅VCoM→=Fz,limbVz,CoM+Fx,limbVx,CoM+Fy,limbVy,CoM,

and the positive components of P_limb_ – for all legs – provide the mechanical work ‘cost’. Note that the instantaneous limb power is the sum of all the components of the right hand side: a positive power due to a vertical force and velocity at the same time as a negative power due to a horizontal force and velocity would cancel; a simple vaulting or pendulum action would be calculated as passive and demand no limb power.”